# Prediction error induced motor contagions in human behaviors

**Tsuyoshi Ikegami[1][†]\*, Gowrishankar Ganesh[2][†], Tatsuya Takeuchi[3], Hiroki Nakamoto[3]\***

[1]Center for Information and Neural Networks, National Institute of Information and Communications Technology, Osaka, Japan; [2]CNRS-AIST JRL (Joint Robotics Laboratory), UMI3218/RL, Intelligent Systems Research Institute, National Institute of Advanced Industrial Science and Technology, Tsukuba, Japan; [3]Faculty of Physical Education, National Institute of Fitness and Sports in Kanoya, Kanoya, Japan

**Abstract** Motor contagions refer to implicit effects on one's actions induced by observed actions. Motor contagions are believed to be induced simply by action observation and cause an observer's action to become similar to the action observed. In contrast, here we report a new motor contagion that is induced only when the observation is accompanied by prediction errors - differences between actions one observes and those he/she predicts or expects. In two experiments, one on whole-body baseball pitching and another on simple arm reaching, we show that the observation of the same action induces distinct motor contagions, depending on whether prediction errors are present or not. In the absence of prediction errors, as in previous reports, participants' actions changed to become similar to the observed action, while in the presence of prediction errors, their actions changed to diverge away from it, suggesting distinct effects of action observation and action prediction on human actions.
DOI: https://doi.org/10.7554/eLife.33392.001

**\*For correspondence:**
ikegami244@gmail.com (TI);
nakamoto@nifs-k.ac.jp (HN)

[†]These authors contributed equally to this work

**Competing interests:** The authors declare that no competing interests exist.

## Introduction

Our motor behaviors are shaped not just by physical interactions (*Shergill et al., 2003*; *Ganesh et al., 2014*; *Takagi et al., 2017*) but also by a variety of perceptual (*Heyes, 2011*; *Chartrand and Bargh, 1999*; *Ikegami and Ganesh, 2014*; *Cook et al., 2012*; *Ganesh and Ikegami, 2016*) interactions with other individuals. *Motor contagions* are the result of such a perceptual interaction. They refer to implicit changes in one's actions caused by the observation of the actions of others (*Blakemore and Frith, 2005*; *Becchio et al., 2007*). Studies over the past two decades have isolated various motor contagions in human behaviors, from the so called automatic imitation (*Brass et al., 2001*; *Heyes, 2011*) and emulation (*Edwards et al., 2003*; *Becchio et al., 2007*; *Gleissner et al., 2000*), to outcome mimicry (*Gray and Beilock, 2011*) and motor mimicry (*Chartrand and Bargh, 1999*; *Chartrand and Baaren, 2009*). These motor contagions are induced simply by action observation and have a signature characteristic - they cause certain features of one's action (such as kinematics [*Brass et al., 2001*; *Heyes, 2011*; *Kilner et al., 2003*], goal [*Edwards et al., 2003*; *Becchio et al., 2007*; *Gleissner et al., 2000*], or outcome [*Gray and Beilock, 2011*]) to become similar to that of the observed action. In contrast, here we report a new motor contagion that is induced not simply by action observation, but when the observation is accompanied by *prediction errors* - differences between actions one observes and those he/she predicts or expects. Furthermore, this contagion may not lead to similarities between observed actions and one's own actions. Here we report results from two experiments to show that distinct motor contagions are induced by the observation of the same actions depending on whether prediction

**eLife digest** Watching sports sometimes causes people to unintentionally move in the same way as the athlete they are observing. This type of unconscious mimicry is called a motor contagion. Observing everyday actions can also trigger motor contagion, and plays an important role in social interactions.

So far, studies have focused on understanding how observing an action leads to motor contagion. They have not factored in the fact that in everyday life individuals consciously or unconsciously predict observed actions by others. Sometimes these predictions are wrong, leading to so called 'prediction errors'. It was not clear whether motor contagion occurs when the viewer has made an incorrect prediction, or if prediction errors change the behavior of the viewer.

Now, Ikegami, Ganesh et al. show that prediction errors influence motor contagion. In one experiment, baseball players were asked to watch a video of an actor pitching a ball toward a target and predict where on the target the ball would hit. Some of the players were given misleading information intended to increase the likelihood they would incorrectly predict where the actor would throw. The players then pitched the ball towards a target themselves. When the players had just watched the actor's throw, their throws became similar to it. When their predictions were wrong, their throws were very different from the actor's throw. The players were not aware of the changes to their throw in either case.

Ikegami, Ganesh et al. also conducted a similar experiment in which other volunteers were asked to observe an actor reaching for a target and then reach for the target themselves. The results were similar: when the volunteers' predictions were wrong, they reached in different ways to the actor.

This may be a new type of motor contagion. Learning more about this effect could help researchers to better understand the adjustments people make to their social behaviors and give new insights about the brain mechanisms that underlie normal human actions and social interactions. Sports trainers or physical therapists might also use this information to develop better strategies for maintaining athlete performances or helping people to recover movement after an injury or illness.
DOI: https://doi.org/10.7554/eLife.33392.002

errors are present or not. We show that these contagions are present not only in high dimensional whole body movement tasks such as baseball pitching (Experiment-1) (*Ikegami et al., 2017*), but also simple day-to-day movement tasks such as arm reaching (Experiment-2).

## Results

### Experiment-1

Thirty varsity baseball players participated in our Experiment-1. The sample size was determined by a power analysis (see Materials and methods). The participants were randomly assigned to one of three groups (n = 10 in each): No prediction error (nPE) group, Prediction error (PE) group, and Control (CON) group. The participant's baseball experience was balanced across the three groups (F $(2,27)$=1.431, p=0.257, $\eta_p^2$=0.096). The participants in the nPE and PE groups performed five *throwing* sessions (*Figure 1A*) that were interspersed with four *observation* sessions (*Figure 1B,C*). The participants in the CON group performed only the throwing sessions. Instead of the observation sessions, they took a break in between the throwing sessions for a time period equivalent to the length of the observation sessions.

In the throwing session, the participants in all the groups threw a baseball aiming for the center of a 'strike-zone' sized square target placed over the 'home plate' (*Figure 1A*). They made their throws while wearing 'occlusion' goggles that turned opaque when the participants released the ball from their hand (see *Figure 1A*). The participants could thus see the target for aiming, but could not see where their ball hit the target. Each throwing session included ten throws.

In the observation session, the participants in the nPE and PE groups watched a video of throws made by an unknown baseball pitcher. After each throw, a numbered grid (see *Figure 1B*) appeared on the target (in the video) once the ball hit the target, and the participants were asked to report the grid number corresponding to where they saw the ball hit the target. The purpose of this

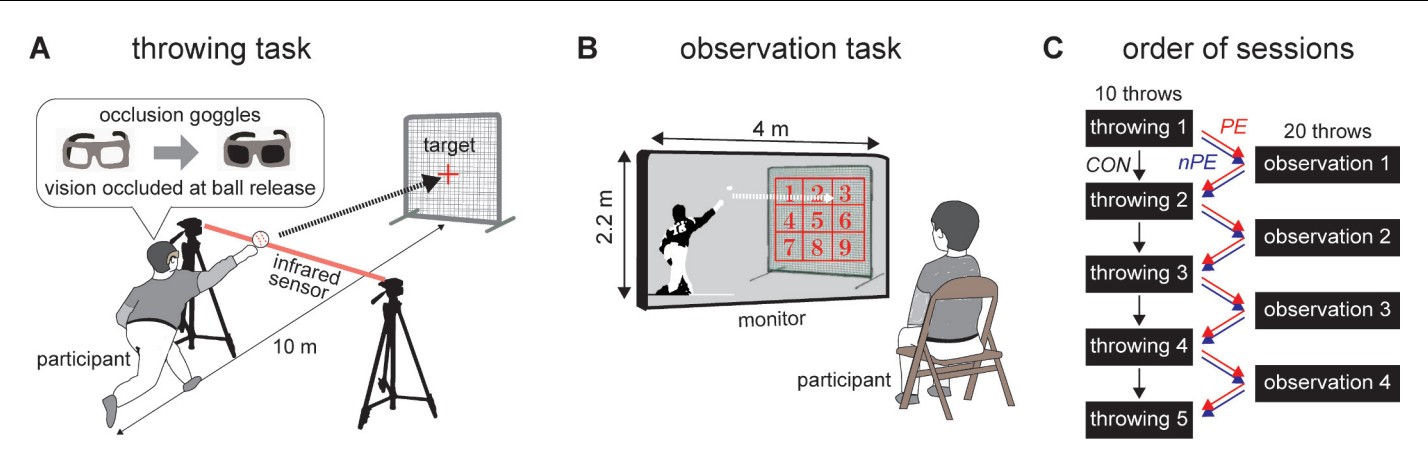

**Figure 1.** Experiment-1 consisted of two tasks. (**A**) In the *throwing task*, the participants threw balls aimed for the center of a target while wearing 'occlusion' goggles, that turned opaque when the participant's arm crossed the infrared beam. This prevented them from seeing where their throw hit the target. (**B**) In the *observation task,* the participants were asked to observe the video of throws made by a baseball pitcher and vocally report a number on a nine part grid corresponding to where the throw hit the target. (**C**) Five throwing sessions were interspersed with four observation sessions for the participants in the nPE group (blue arrow) and PE group (red arrow), while the participants in the CON group (black arrow) performed only the throwing sessions, but took a break (equivalent to the length of an observation session) between the throwing sessions.

DOI: https://doi.org/10.7554/eLife.33392.003

reporting task was to ensure that the participants maintained their attention on the target in the video. To cancel out any spatial bias with respect to the observed actions, half participants in each group (called upper-right observing participants) were shown throws that predominantly hit the upper-right corner of the target (most frequent at #3, see yellow gradient *Figure 2B* and Materials and methods for observed throw distribution). The other five participants (called lower-left observing participants) were shown a video of throws predominantly hitting the lower-left corner of the target (most frequent at #7, see Materials and methods). Each observation session included 20 observed throws.

Different instructions were provided to the nPE and PE groups to manipulate the prediction errors induced in them. The participants in the nPE group were told that 'the pitcher in the video is aiming for different grid numbers on the target across trials. These numbers were provided by the experimenter and we display only those trials in which he was successful in hitting the number he aimed for'. On the other hand, participants in the PE group were instructed that 'the pitcher in the video is aiming for the center of the target'.

The two different instructions were designed to lead to greater prediction errors in the PE group compared with the nPE group. The instruction to the nPE group prevents the participants from having any prior expectation of the outcome of an observed throw, and was thus expected to attenuate any prediction error. In contrast, the instruction to the PE group makes the participants expect the observed throws to hit near the target center. Similar to previous studies (*Ikegami and Ganesh, 2014*; *Ondobaka et al., 2015*), this instruction was thus expected to induce a difference between the throw outcome expected by the participants and the actual outcome observed by them. Specifically, we expected the upper-right and lower-left observation participants in the PE group to experience prediction errors, directed towards the upper-right and lower-left, respectively.

The participants' task performance in the throwing session was evaluated as a change in the throw hit location. The position of the hit locations was measured along the 'parallel' and 'orthogonal' diagonals (see green arrows in *Figure 2A*). The diagonal joining the upper-right and lower-left corners, that were the predominant locations of the pitcher's throws in the observed video, was named the 'parallel' diagonal. Data from the upper-right and lower-left observing participants were analyzed together by flipping the coordinate of the data from the lower-left observing participants (see Materials and methods).

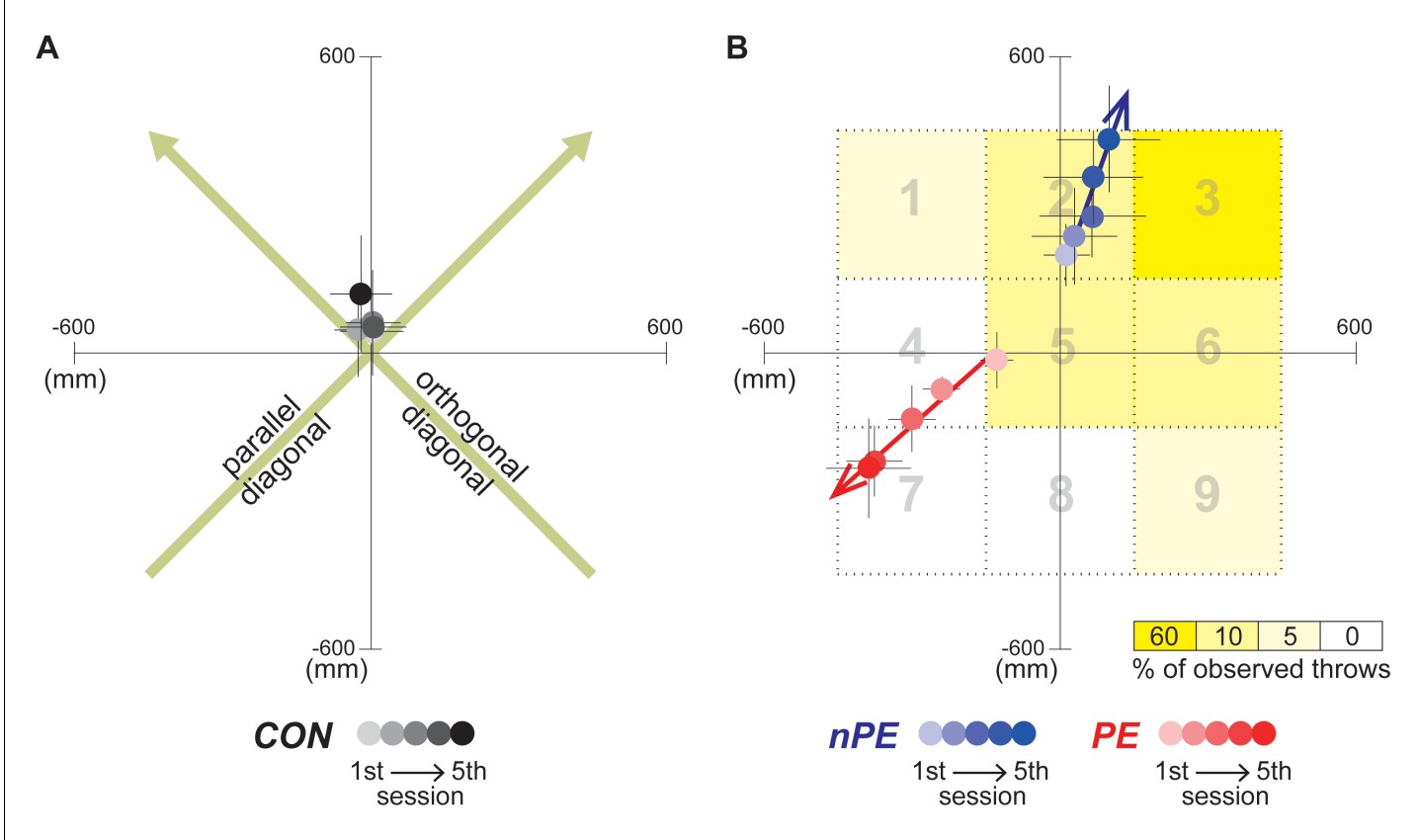

**Figure 2.** Throw hit locations of participants in Experiment-1. (**A**) The across participant average (and s.e.) of hit locations in the CON group shown in gradients of gray, with a darker color representing a later throwing session. The green arrows show the 'parallel' and 'orthogonal' diagonals, which are used as the reference coordinates for the data plotting and quantification analysis for the nPE and PE groups. (**B**) The across participant average (and s. e.) of hit locations in the nPE group (blue data) and the PE group (red data). The grid area markings shown to the participants after each observed throw, are shown in gray. The yellow gradient indicates the percentage of the observed throws hitting each grid area in the observation sessions for the *upper-right* observing participants, who were shown the pitcher's throws biased to the upper-right corner of the target. Note that the hit locations were plotted by flipping the data from the *lower-left* observing participants along the parallel and orthogonal diagonals, such that the predominant direction of pitcher's throws observed by all participants in both the PE and nPE groups were toward the *upper-right* corner of the target. The nPE groups tend to progressively deviate towards the observed throws, while the PE groups tend to deviate away from the observed throws. The colored arrows indicate the linear fit of the participant-averaged hit locations across the throwing sessions.
DOI: https://doi.org/10.7554/eLife.33392.004

First, in the observation session, the accuracy (% correct) of the report was comparable between the two groups (nPE: 96.25 ± 1.48 (mean ± s.d.) %, PE: 95.63 ± 3.32 %; two sample t-test, t(18) =0.516, p=0.612). This ensures that the level of attention to the video was similar between the two groups.

The performance in the throwing session, however, dramatically differed between the two groups. The throws by the nPE group progressively drifted towards where the pitcher in the video predominantly threw the ball (blue data in *Figure 2B*). This pattern is similar to the motor contagion reported previously as outcome mimicry (*Gray and Beilock, 2011*). In contrast, the throws by the PE group progressively drifted *away* from where the pitcher in the video predominantly threw the ball (red data in *Figure 2B*). The hit locations along the parallel diagonal (*Figure 3*) showed a significant interaction between the sessions and groups (F(8,108)=5.124, p=2 × $10^{-5}$, $\eta_p^2$=0.275). Across the sessions, the throws in the nPE group (blue data in *Figures 2B* and *3*) significantly drifted towards the direction of observed pitcher's throws (F(4,36)=2.910, p=0.035, $\eta_p^2$=0.244; first vs fifth sessions by Tukey's test: p=0.034) while the throws in the PE group (red data in *Figures 2B* and *3*) significantly drifted away from the observed pitcher's throws (F(4,36)=5.170, p=2 × $10^{-3}$, $\eta_p^2$=0.365; first vs fifth sessions by Tukey's test: p=5 × $10^{-3}$). The throws in the CON group (black data in

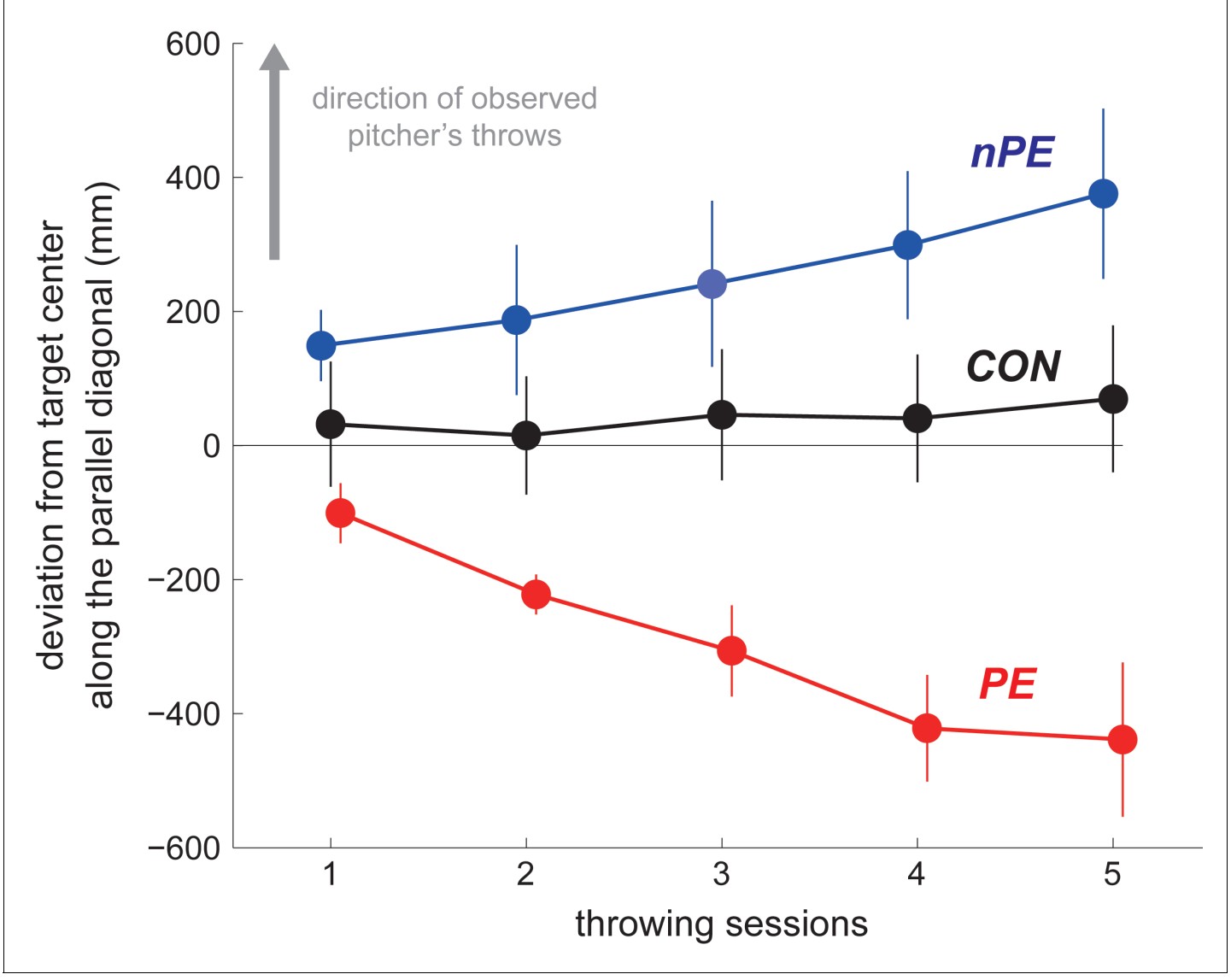

**Figure 3.** Changes in the hit locations of the participants' throws across the throwing sessions in Experiment-1. The participant-averaged deviations from the target center along the parallel diagonal were plotted by flipping the data from the lower-left observing participants, such that the directions of pitcher's throws observed by all participants in both the nPE and PE groups were in the positive ordinate (indicated by gray arrow). The data from the CON, nPE, and PE groups are plotted in black, blue, and red, respectively. The throws by the nPE group progressively deviated towards where the pitcher in the video threw the ball most frequently. In contrast, the throws by the nPE group progressively deviated away from where the pitcher in the video threw the ball most frequently. The throws by the CON group did not change across sessions. Error bars indicate standard error.
DOI: https://doi.org/10.7554/eLife.33392.005

*Figures 2A* and *3*) did not show such a drift ($F_{(4,36)}=0.297$, $p=0.878$, $\eta_p^2=0.032$) along the parallel diagonal.

On the other hand, the hit locations along the orthogonal diagonal in all the nPE, PE, and CON groups showed no significant differences across the throwing sessions ($F_{(4,108)}=1.762$, $p=0.142$, $\eta_p^2=0.061$) and between the groups ($F_{(2,27)}=1.150$, $p=0.332$, $\eta_p^2=0.079$). This result confirms that the drifts observed in the nPE and PE groups are not the result of possible cognitive fatigue induced by the extra observation task they performed (compared with the CON group), or cognitive biases induced by the validity of the instructions given to them. In either case, we would have expected their throws to also drift in directions other than along the parallel diagonal. The focused drifts along the parallel diagonal suggest that the drifts were induced by the bias in the observed pitcher's

throws (in the nPE group) and the prediction errors (in the PE group), both of which were present specifically along the parallel diagonal.

In addition, the effects of observed actions on the participants' actions emerged as an increase in spatial bias but not in spatial variability in their action outcome. For the orthogonal diagonal, the within-participant variability in their hit locations, measured by variance in each throwing session (see Materials and methods), showed no significant differences across the sessions in all the three groups (Friedman' test, nPE: $\chi^2(4)=7.44$, p=0.114; PE: $\chi^2(4)=0.32$, p=0.989; CON: $\chi^2(4)=3.2$, p=0.525). For the parallel diagonal, although the PE groups showed a significant change in variance (nPE: $\chi^2(4)$ =3.28, p=0.512; PE: $\chi^2(4)=10.48$, p=0.033; CON: $\chi^2(4)=3.2$, p=0.525), we did not observe any clear trend in the median values of the within-participant variability across the sessions (first session: 7.772; second: 5.360; third: 7.092; fourth: 7.008; fifth: 7.767 $\times 10^4$ mm$^2$, respectively). And, a post hoc analysis found no significant difference between any pair of sessions (Wilcoxon signed rank test, Zs <1.886, ps >0.059, which was considerably above the Bonferroni corrected significance level of p=0.005).

Together, these results clearly show that the observation of a same action can lead to distinct motor contagions depending on whether the observation takes place in the presence or absence of prediction errors.

## Experiment-2

Next, to check whether this prediction error induced contagion is specific to sports experts, and to verify whether it can also be observed in simple everyday movement tasks, we conducted a second follow-up experiment in which we used a similar experimental design to before but tested average adult participants in an arm reaching task (see Materials and methods).

Thirty right-handed averaged male participants were randomly assigned to one of three groups (n = 10 in each): nPE, PE, and CON groups. The participants in the nPE and PE groups performed five *reaching* sessions and four *observation* sessions (*Figure 4A*). The participants in the CON group performed only the reaching sessions.

In the reaching sessions, the participants made right arm reaching movements toward a touch screen to touch the center line, among three vertical lines presented on the screen, with their index fingers (*Figure 4A*). They again wore occlusion goggles which, similar to the throwing experiment, became opaque as soon as their arm movements started, preventing them from observing where their finger touched the screen. Each reaching session included ten reaches.

In the observation session, the participants in the nPE and PE groups watched a video of an unknown individual (henceforth, the *actor*) reaching towards the same touch screen (*Figure 4A*). After each reach, the participants were asked to report if the actor had touched the 'right' (area between the center and right lines), 'left' (area between the center and left lines), or 'outside' (of the right and left lines). Half the participants in each group watched a video showing the actor predominantly touching the right area (called right observing participants), and the other half watched a video showing the actor predominantly touching the left area (called left observing participants), respectively (see Materials and methods). Each observation session included 20 observed reaches.

To manipulate the prediction errors, we again provided different instructions to the nPE and PE groups. For the nPE group, the right and left observing participants were told that 'the actor in the video is aiming for the right area' and 'the actor in the video is aiming for the left area', respectively. This instruction was expected to minimize the difference between the actor's touch location expected by the participants, and the actual location observed by them, thus leading to suppression in prediction errors. On the other hand, participants in the PE group were instructed that 'the actor in the video is aiming for the center line'. This instruction was expected to induce substantial prediction errors as in Experiment-1.

To evaluate the participants' reaching performance in the reaching session, the touch locations were measured along the x-axis on the screen. Similar to the main experiment, data from the right and left observing participants were analyzed together by flipping the coordinate of the data from the left observing participants (see Materials and methods).

First, in the observation session, we confirmed that the accuracy (% correct) of the report was comparable between the two groups (nPE: 93.20 ± 3.94 (mean ± s.d.) %, PE: 93.00 ± 5.10 %; two sample t-test, t(18)=0.098, p=0.923). Next, in the reaching session, we observed similar results to Experiment-1; the touch locations by the participants (*Figure 4B*) showed significant interaction

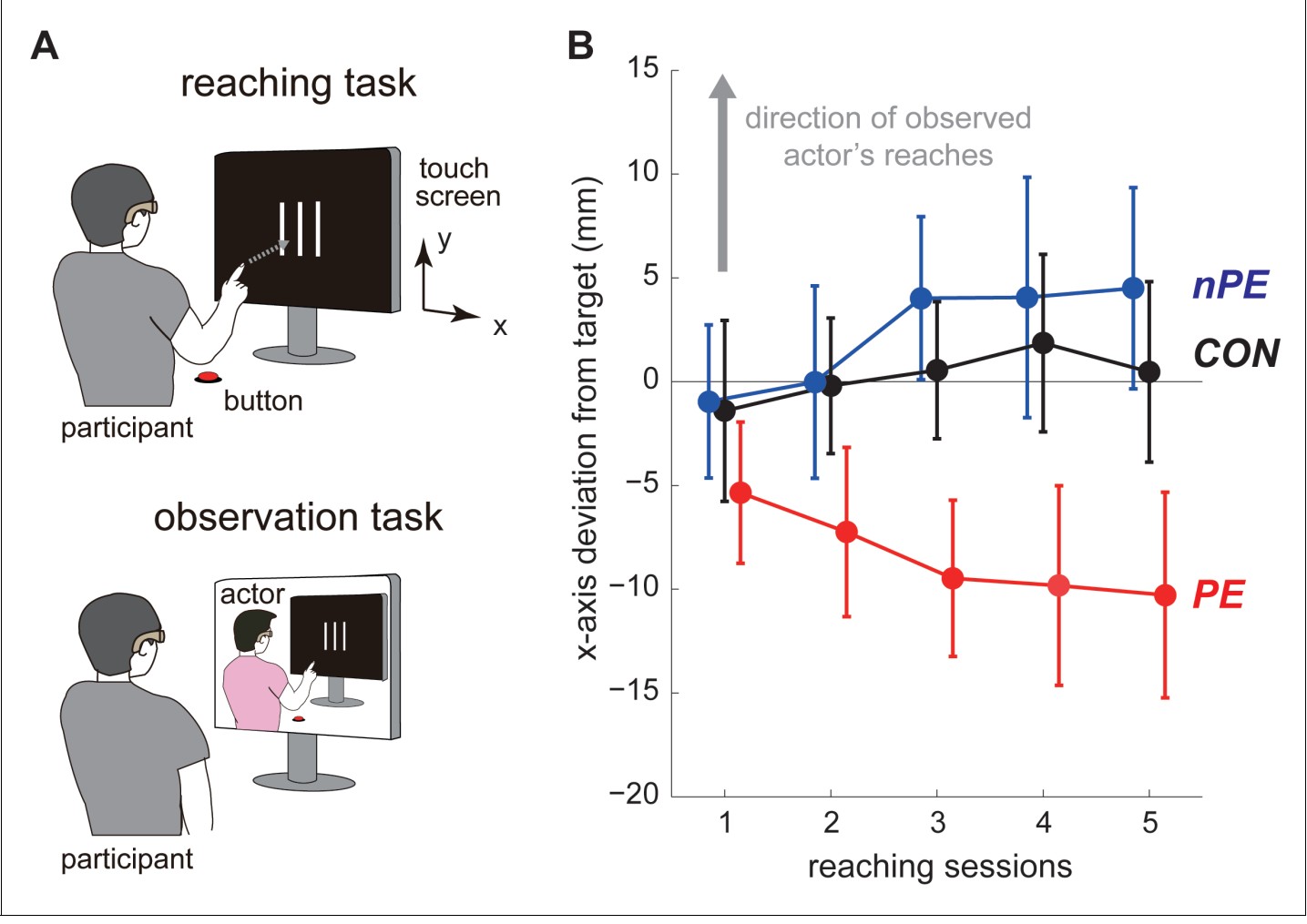

**Figure 4.** Experiment-2 consisted of two tasks similar to Experiment-1. (A) In the *reaching task,* the participants made reaching movements to touch the center line (among the three vertical lines) on a touch screen with their index fingers. They wore 'occlusion' goggles that turned opaque when the participant started to reach. This prevented them from seeing where their index finger touched the screen. In the *observation task,* with their occlusion goggles open, the participants were asked to observe the video of reaches made by an actor and report where the actor touched the screen. (B) Changes in the participants' touch locations across the throwing sessions. The participant-averaged deviations from the center line along the x-axis were plotted by flipping the data from the left observing participants, such that the directions of the actor's reaches observed by all participants in both the nPE and PE groups were in the positive ordinate (indicated by gray arrow). The data from the CON, nPE, and PE groups are plotted in black, blue, and red, respectively. The touch locations by the nPE group progressively deviated towards where the actor touched most frequently on the screen. In contrast, the touch locations by the PE group progressively deviated away from where the actor touched most frequently on the screen. The touch locations by the CON group did not change across sessions. Error bars indicate standard error.

DOI: https://doi.org/10.7554/eLife.33392.006

The following figure supplement is available for figure 4:

**Figure supplement 1.** Touch location in Experiment-2.

DOI: https://doi.org/10.7554/eLife.33392.007

between the sessions and groups (F(8,108)=2.568, p=0.013, $\eta_p^2$=0.160). Across the sessions, the touch locations by the nPE group (blue data in *Figure 4B*) drifted towards the direction of observed actor's touch locations (F(4,36)=2.307, p=0.077, $\eta_p^2$=0.204), although it marginally missed statistical significance. More importantly for this study, the touch locations by the PE group (red data in *Figure 4B*) significantly drifted away from the direction of the observed actor's touch locations (F(4,36)=2.683, p=0.047, $\eta_p^2$=0.230). The touch locations in the CON group (black data in *Figure 4B*) did not show any substantial change (F(4,36)=0.639, p=0.638, $\eta_p^2$=0.066). In addition, the within-

participant variability in their touch locations showed no significant differences across the reaching sessions (nPE: $\chi^2(4)=5.04$, p=0.283; PE $\chi^2(4)=4.80$, p=0.308; CON: $\chi^2(4)=9.44$, p=0.051).

Thus, Experiment-2 successfully reproduced the Experiment-1 results, which suggests that the two distinct motor contagions can affect even basic everyday movements like arm reaching. Crucially, the clear motor performance changes in the PE groups observed in the two experiments validate the presence of a new motor contagion which is induced by prediction errors during action observation.

## Discussion

Previous motor contagion studies have extensively examined how observation of the actions of others affects action production (*Heyes, 2011*; *Chartrand and Bargh, 1999*; *Becchio et al., 2007*; *Brass et al., 2001*; *Edwards et al., 2003*; *Gray and Beilock, 2011*; *Kilner et al., 2003*). On the other hand, while previous action prediction studies have examined how action production (*Mulligan et al., 2016*; *Hamilton et al., 2004*), or the ability to produce an action (*Knoblich and Flach, 2001*; *Urgesi et al., 2012*; *Kanakogi and Itakura, 2011*) affects prediction of observed actions, the converse, the effect of action prediction on one's actions, has rarely been examined. One exception is our previous study (*Ikegami and Ganesh, 2014*), which reported that an improvement in the ability to predict observed actions affects the observer's ability to produce the same action. The observations from this previous study are likely to be a prediction error induced motor contagion, but as the direction of the observed actions and the corresponding prediction errors were not controlled in the study, it was difficult to conclude this cause. Overall, the effects of action prediction and action observation on action production have never been compared. This study compares the two effects for the first time by modulating the prediction errors perceived by participants, when they observe throws or reaches by another individual. We show that, while in the absence of prediction errors (nPE group), the action observation makes the participants' actions (throws and reaches) become similar to the observed actions (as in previous reports [*Heyes, 2011*; *Gray and Beilock, 2011*; *Blakemore and Frith, 2005*]), in the presence of prediction errors (PE group), the same action observation makes the participants' actions diverge away from the observed actions (*Figures 2B*, *3* and *4B*). Our results thus suggest the presence of a distinct, prediction error induced motor contagion in human behaviors.

The previously reported motor contagions are believed to be caused by the presence of 'long-term' sensorimotor associations (*Heyes, 2011*; *Cook et al., 2014*), in which an observed action automatically activates an individual's sensorimotor representations of the same action, making his/her actions similar to the observed actions. While further studies are required to understand the mechanisms underlying the prediction error induced motor contagion, it is still interesting to note the parallel between this motor contagion and behavioral characteristic of motor learning. When learning a new motor task, individuals are believed to utilize prediction errors from self-generated actions to change their sensorimotor associations, and correct their actions (*Shadmehr et al., 2010*). The motor contagion in the PE group seems to cause participants to make similar action corrections to compensate for their prediction errors, strangely even though the prediction errors are from actions made by another (observed) individual, rather than from self-generated actions. Therefore, while the previously reported motor contagions may be explained as a facilitation/interference in action production by the sensorimotor associations (*Heyes, 2011*; *Cook et al., 2014*) activated by observed actions, the new contagion we present here is probably a result of erroneous changes in the sensorimotor associations, caused by the observed prediction errors (*Ikegami and Ganesh, 2017*).

The prediction error induced contagions presented here seem similar to the phenomena of observational motor learning, in which the observation of an action is reported to aid subsequent motor learning (*Adams, 1986*; *Schmidt and Lee, 2005*; *Mattar and Gribble, 2005*; *Buckingham et al., 2014*). It has been suggested that observational motor learning is possible because of an observation induced adaptation of the *inverse model* (or controller) that determines motor commands for a given task (*Brown et al., 2010*). On the other hand, in a recent study (*Ikegami and Ganesh, 2017*), we showed that the behavioral changes resulting from prediction error induced contagions can be explained only by an influence in an individual's *forward model,* that uses the motor commands to estimate one's movement outcomes. This difference in effects could result from two factors. First, unlike observational motor learning (*Maslovat et al., 2010*;

*Ong et al., 2012*), our previous (*Ikegami and Ganesh, 2014*; *Ikegami and Ganesh, 2017*) and present studies did not require the participants to learn a new motor task, and in fact included participants who were arguably over-trained in the task. This strongly suggests that the contagion effects in these studies are implicit (not willed by the participants). And second, unlike the observational motor learning studies, this study systematically manipulates the participants' belief regarding the observed actions to control the direction of prediction errors. The lack of an explicitly required learning and the effect of belief are probably key to whether and to what extent, observed actions affect the inverse, or forward modeling process, or both. However, further studies are required to address this question concretely. Finally, further studies are also needed to examine whether the prediction error induced motor contagions play a functional role in our motor actions. If the contagions are proved to contribute to motor skill acquisition or motor adaptation to a new environment, we may need to re-define the phenomenon as a new learning process rather than a new motor contagion.

In conclusion, our results clearly show that action observation and action prediction can induce distinct effects on an observer's actions. Understanding the interactions between these distinct effects is arguably critical for the complete understanding of human skill learning. For example, it can enable a better understanding of the link between the mechanisms of motor contagions and motor learning, and help develop better procedures to improve motor performances in sports and rehabilitation.

## Materials and methods

### Participants
#### Experiment-1
Thirty right-handed male varsity baseball players (20.33 ± 1.62 (mean ± s.d.) years old) with normal or corrected vision, took part in Experiment-1. Their years of experience in playing baseball were 11.43 ± 1.98 (mean ± s.d.). They were randomly assigned to one of three groups of ten participants each: the No prediction error (nPE) group, the Prediction error (PE) group, and the Control (CON) group. All participants gave informed consent before the experiment and the experiment was approved by the ethics committee in National Institute of Fitness and Sports in Kanoya, and conducted according to the Declaration of Helsinki.

The sample size was determined by a power analysis using G*power (*Faul et al., 2007*) (Repeated measure ANOVA within-between interaction, $\alpha = 0.05$, $\beta = 0.80$, $\eta_p^2 = 0.06$ (medium value), correlation among repeated measures = 0.5, non-sphericity correction = 1). The power analysis provides the sample size of n = 9 for each group. We determined our sample size of n = 10 as a more conservative choice with respect to a type-1 error.

#### Experiment-2
Thirty male participants (22.43 ± 1.79 (mean ± s.d.) years old) with normal or corrected vision, took part in this study. The sample size was determined based on Experiment-1 because the experimental design was same (in terms of the numbers of participant groups, motor task sessions, and observation sessions). All participants were right-handed (laterality quotient, 87.59 ± 17.60 (mean ± s.d.)) according to the Edinburgh Inventory (*Oldfield, 1971*). The participants were randomly assigned to one of three groups (n = 10 in each): nPE, PE, and CON groups. This experiment was conducted according to the principles in the Declaration of Helsinki. All participants gave informed consent before the experiment, and the experiments were approved by the ethics committee of the National Institute of Information and Communications Technology.

### Task and apparatus
#### Task and apparatus in Experiment-1
All experiments were performed in an indoor baseball facility. The participants in the nPE and PE groups performed in alternating *throwing* (five sessions), and *observation* (four sessions) sessions (*Figure 1C*), while the participants in the CON group performed only five throwing sessions.

## Throwing sessions

In the throwing sessions, participants in all the groups were required to throw balls aimed at the center of a target placed on the 'home plate' 10 m away. The target was a 'strike zone' sized square (0.9 × 0.9 m) and its center was indicated by a cross (*Figure 1A*). The participants wore liquid-crystal shutter goggles (PLATO, Translucent Technologies, Toronto) during the throwing sessions. The goggles allowed the participants to see the target when taking aim but occluded their vision for 3 s, starting immediately after they released the ball from their hand for each throw. The timing of ball release was detected by an infrared transmitter-sensor system (AO-S1, Applied Office, Tokyo), which sends a TTL signal to the goggles when the participants' hand intersects the infrared beam between the transmitter and sensor (*Figure 1A*). The position of the infrared transmitter-sensor setup was adjusted for each participant such that the goggle was shut at the timing of their ball release. This system ensures that the participants do not see their ball flight trajectory and the hit location on the target, preventing them from using visual feedback to correct subsequent throws. Each throwing session included ten throws. The ball hit locations were recorded by a digital video camera at 60 fps (Sony HDR-CX560).

## Observation sessions

In the observation sessions, the participants (in the nPE and PE groups) sat on a chair 15 m from a large monitor (2.2 × 4.0 m) and watched a life-size video of a right-handed baseball pitcher throwing a ball. Unknown to the participants, the pitcher in the video was asked to aim for various areas on the target indicated by an experimenter (see next subsection on observation session videos), and we pre-selected videos to each participant. For five participants in each group, the spatial distribution of throws by the pitcher in the videos was biased to the upper right corner of the target (see *Figure 2B*): 12 balls (60 %) in #3; two balls (10 %) in each of #2, #5, #6; one ball (5 %) in each of #1, #9. For the other five participants in each group, the distribution of throws in the videos were biased to the lower left corner of the target (see right panel in *Figure 2B*): 12 balls in #7; two balls in each of #4, #5, #8; one ball in each of #1, #9.

Each observation session included 20 video clips and lasted about three minutes. Nine numbered grid areas (0.3 × 0.3 m for each grid) appeared on the target immediately after each throw hit the target and then the participants were asked to vocally announce the grid number corresponding to where the throw hit the target. The participants' answers were recorded by an experimenter. The task helped us ensure that the participants maintained concentration in the task and watched each throw.

The prediction errors during the observation session were manipulated by a difference of instructions between the nPE and PE groups, provided before each observation session. The participants in the PE group were told that 'the pitcher in the video is aiming for the center of the target'. This instruction was expected to generate prediction errors between what the participants expect and what they see in the video. On the other hand, the participants in the nPE group were told that 'the pitcher in the video is aiming for various grid numbers on targets. These numbers were provided by the experimenter and we display only those trials in which he was successful in hitting the number he aimed for'. This instruction was expected to prevent the participants from expecting a throw outcome, and hence attenuate prediction errors. The participants in the CON group did not watch a video (they did not have observation sessions), but instead sat on the chair and took a break for three minutes, equivalent to the length of the observation session. The behavior of the CON group was used to check for possible drifts in the participants' throws because of the persistent lack of feedback in the throwing task.

## Observation session videos

To create the video clips for the observation sessions, we recorded movies of a pitcher throwing balls toward the target 10 m away. We used a video camera (Sony HDR-CX560, recoding at 60 fps). The camera was placed diagonally (relative to the pitcher–target line) behind the pitcher, 3 m distance away, and recorded the pitcher's kinematics.

The pitcher was asked to aim his throws to each of the nine areas (#1 to #9) on the target (shown in *Figure 1B*). He continued to aim for the same target area until he hit it ten times. This procedure provided us with video clips of ten successful throws to each of the nine target areas. From these clips, we chose two clips for throws to #2, #5, #6, (or #4, #5, #8) and one clip for a throw to #1 and

#9, and six clips each to throws to #3 (or #7) based on the visibility of ball release and hit location. The six clips (to #3 and #7) were used twice in each observation session.

Next, we edited the selected video clips with Adobe Premiere Pro CS6. Each video clip was temporally clipped from between 2000 ms before ball release and the moment of the target impact. For 3000 ms after the target impact, a grid showing the nine numbered grid areas (*Figure 1B*) was superimposed on the target area in the video. The order of presentation of video clips during observation task was randomized among sessions and participants.

## Task and apparatus in Experiment-2

As in Experiment-1, the participants in the nPE and PE groups performed in alternating reaching (five sessions) and four observation sessions (*Figure 4*), while the participants in the CON group performed only five reaching sessions.

### Reaching sessions

In the reaching sessions, the participants sat on a chair facing a 19 inch LCD screen (ET1915L-8CJA-1-BG-G, Tyco Electronics) placed 45 cm away from their eyes. The participants in all the groups were required to make a reaching movement and touch the center line among three 8 cm vertical lines presented on the screen with their right index finger. The three lines were positioned at the center of the screen and separated by 2.5 cm (*Figure 4A*). As in Experiment-1, the participants wore liquid-crystal shutter goggles (T.K.K.2275, Takei Scientific Instruments, Niigata) during the reaching sessions. The shutter was opened when the participants pressed a round button (6.5 mm diameter). The participants were required to press the center of the button with their right index finger. The button was positioned 15 cm ahead of the eye in the midsagittal plane. The goggles allowed the participants to see the target when taking aim but occluded their vision immediately after they released the button to reach for the screen. This system ensured that the participants do not see their touch location on the screen, preventing them from using visual feedback to correct subsequent reaches. Each reaching session included ten reaches. Their touch locations were detected with a touch sensor.

### Observation sessions

In the observation sessions, the participants (in the nPE and PE groups) watched a video of an actor making a reaching movement, which was presented on the same screen used in the reaching session. The actor in the video wore the shutter goggles and made a reaching movement toward the three vertical lines on the screen as the participants performed in the reaching sessions. Unknown to the participants, the actor in the video was asked to aim for various areas between the right and left vertical lines on the screen. We pre-selected videos for each participant based on the reached areas, which are 10 segmented target areas between the right and left vertical lines with intervals of 5 mm (#1 to #10, see *Figure 4—figure supplement 1*). For five participants in each group, the spatial distributions of the actor's reaches in the videos were biased to the right area between the right and center vertical lines (*Figure 4—figure supplement 1*): eight reaches (40 %) in #8; three reaches (15 %) in each of #6, #7; two reaches (10 %) in each of #5, #6, #10. For the other five participants in each group, the distributions of throws in the videos were biased to the left area between the left and center vertical lines (see *Figure 4—figure supplement 1*): eight reaches (40 %) in #3; three reaches (15 %) in each of #2, #4; two reaches (10 %) in each of #1, #5, #6.

Each observation session included 20 video clips and lasted about two minutes. The participants were instructed to indicate where the actor touched the screen by writing down one of three possible answers at every trial: 'right', 'left', or 'out'. The prticipants were asked to answer as 'right' or 'left' when the actor's touch was seen to be between the center and right vertical lines, or between the center and left lines, respectively. They were asked to answer as 'out' when the touch was seen to be beyond the right and left lines.

The prediction errors during the observation session were manipulated by a difference of instructions between the nPE and PE groups, provided before each observation session. The participants in the nPE group were told that 'the actor in the video is aiming for the right (or left) area between the center and right (or left) lines'. This instruction was expected to make the participants expect the actor to touch the right (or left) area, and thus attenuate a difference between the actor's touch location expected by the participants and the actual location observed by them, leading to suppression

in prediction errors. On the other hand, the participants in the PE group were told that 'the actor in the video is aiming for the center line among the three vertical lines'. This instruction was expected to generate prediction errors between what the participants expect and what they observe in the video. Specifically, we expected the right and left observation participants in the PE group to experience prediction errors, directed towards the right and left, respectively. The participants in the CON group did not watch a video (they did not have observation sessions) but instead sat on the chair and took a break for two minutes, equivalent to the length of the observation session.

## Observation session videos

To create the video clips for the observation sessions in Experiment-2, we recorded movies of an actor making reaching movements toward the screen wearing shutter goggles. We used a video camera (Casio EX-100F, recoding at 30 fps). The camera was placed diagonally right behind the actor, 50 cm distance away, and recorded the actor's kinematics.

The actor was asked to make reaching movements aiming for various locations between the right and left vertical lines on the screen. The actor made 150 reaches to the 10 target areas. From these clips, we chose 20 clips for each of the left and right observing participants. The video for the left observing participants includes two clips for reaches to #1, #5, #6 and three clips for reaches to #2 and #4, and eight clips each to throws to #3. The video for the right observing participants includes two clips for reaches to #5, #6, #10 and three clips for reaches to #7 and #9, and eight clips each to throws to #8.

Next, we edited the selected video clips with Adobe Premiere Pro CS6. Each video clip was temporally clipped from ~1000 ms before button release to ~1000 ms after the moment of the screen touch. Immediately after the screen touch, the three vertical lines (*Figure 4A*) disappeared from the screen in the video. The order of presentation of video clips during observation task was randomized among sessions.

## Data analysis

### Hit location analysis in Experiment-1

The hit locations recorded by the camera were digitized using Dartfish (Dartfish, Tokyo, Japan). The hit locations of each throw were first measured in the x-y coordinates where the center of the target was taken as the origin. The throwing task performance in each session was evaluated as the distance of hit location from the target center, averaged over the ten throws for each participant. This value was then averaged across the participants and plotted in *Figure 2A,B*. For statistical analysis, the hit locations by all three groups were analyzed along the diagonals (green arrows in *Figure 2A*), parallel and orthogonal to the line joining the corners of the target where the observed throws predominantly hit (#3 or #7). To pool the participants in one group, the hit locations of the throws by the participants who viewed the pitcher aiming for the lower left corner of the target were flipped along the parallel and orthogonal diagonals about the data in the first throwing session. Note that the hit locations in the first throwing session need not be corrected because these represent the default performance by the participant, before the first observation session. To examine spatial biases in the hit location, we separately performed two-way ANOVAs (three groups × five throwing sessions) on the hit locations of the participants' throws along each of the parallel and orthogonal diagonal. Post hoc pairwise comparisons were performed using Tukey's method. To examine within-participant variability changes in the hit location across the five throwing sessions, we separately performed Friedman's test on variances of the hit locations within each session (non-normal data) along each of the parallel and orthogonal diagonals for each group. Post hoc pairwise comparisons were performed using the Wilcoxon signed rank test.

### Observation performance analysis in Experiment-1

In the observation session, participants in the nPE and PE groups were required to answer where the balls thrown by the pitchers hit the target by orally reporting one of nine possible areas in each trial. The percentage of correct answers in each observation session was calculated for each participant.

## Touch location analysis in Experiment-2

The touch locations of each reach were first measured in the x-y coordinates with the center of the screen as the origin. The reaching task performance in each session was evaluated as the distance of the touch location from the center vertical line along the x-axis, averaged over the ten reaches by each participant. This value was then averaged across the participants and plotted in *Figure 4B*. For statistical analysis, to pool the participants in one group, the touch locations of the reaches by the participants who viewed the actor aiming for the left area were flipped along the x-axis about the data in the first reaching session, similar to Experiment-1. We separately performed two-way ANOVAs (three groups × five throwing sessions) on the touch locations of the participants' reaches. We separately performed Friedman's test on variances of the touch locations within each session (non-normal data) for each group.

## Observation performance analysis in Experiment-2

In the observation session, participants in the nPE and PE groups were required to answer where the actor touched the screen by writing down one of three possible answers in each trial. The percentage of correct answers in each observation session was calculated for each participant.

### Data availability

Data to generate *Figures 2A, B*, *3* and *4B* are available from the Dryad Digital Repository: http://datadryad.org/review?doi=doi:10.5061/dryad.3563k

# Acknowledgements

We thank Mr. Kenta Yamamoto for his help in conducting Experiment-1. We also thank Mr. Satoshi Tada and Mr. Ryo Iwai for their assistance in developing the experimental setup, and Ms. Naoko Katagiri for her help in recruiting participants for Experiment-2. We also thank Dr. Masaya Hirashima and Dr. Nobuhiro Hagura for their valuable comments on a former version of the manuscript. This work was partially supported by JSPS KAKENHI Grant #16H05916 and #26120003 (to TI), #16K12999 and #26702025 (to HN), and #15616710 and #13380602 (to GG).

# Additional information

### Funding

| Funder | Grant reference number | Author |
| --- | --- | --- |
| Japan Society for the Promotion of Science | #16H05916 | Tsuyoshi Ikegami |
| Japan Society for the Promotion of Science | #26120003 | Tsuyoshi Ikegami |
| Japan Society for the Promotion of Science | #15616710 | Gowrishankar Ganesh |
| Japan Society for the Promotion of Science | #13380602 | Gowrishankar Ganesh |
| Japan Society for the Promotion of Science | #16K12999 | Hiroki Nakamoto |
| Japan Society for the Promotion of Science | #26702025 | Hiroki Nakamoto |

The funders had no role in study design, data collection and interpretation, or the decision to submit the work for publication.

### Author contributions

Tsuyoshi Ikegami, Conceptualization, Data curation, Formal analysis, Funding acquisition, Validation, Investigation, Visualization, Methodology, Writing—original draft, Project administration, Writing—review and editing; Gowrishankar Ganesh, Conceptualization, Resources, Funding acquisition, Validation, Investigation, Visualization, Methodology, Writing—original draft, Project administration, Writing—review and editing; Tatsuya Takeuchi, Data curation, Formal analysis, Validation,

Visualization, Project administration; Hiroki Nakamoto, Conceptualization, Resources, Data curation, Formal analysis, Funding acquisition, Validation, Investigation, Visualization, Methodology, Writing—original draft, Project administration, Writing—review and editing

### Author ORCIDs
Tsuyoshi Ikegami ⬚ http://orcid.org/0000-0002-8430-8226

### Ethics

Human subjects: All participants were informed of the experimental procedures in advance and gave informed consent before the experiment. All experiments were performed according to the Declaration of Helsinki. Experiment-1 and Experiment-2 were approved by the ethics committee in National Institute of Fitness and Sports in Kanoya, and the ethics committee in National Institute of Information and Communications Technology, respectively

### Decision letter and Author response
Decision letter https://doi.org/10.7554/eLife.33392.015
Author response https://doi.org/10.7554/eLife.33392.016

## Additional files

### Supplementary files
• Transparent reporting form
DOI: https://doi.org/10.7554/eLife.33392.008

### Data availability
Source data files have been provided for Figures 2A, 2B, 3 and 4B.

The following dataset was generated:

| Author(s) | Year | Dataset title | Dataset URL | Database, license, and accessibility information |
|---|---|---|---|---|
| Tsuyoshi Ikegami, Gowrishankar Ganesh, Tatsuya Takeuchi, Hiroki Nakamoto | 2017 | Data from: Prediction error induced motor contagions in human behaviors | http://dx.doi.org/10.5061/dryad.3563k | Available at Dryad Digital Repository under a CC0 Public Domain Dedication |

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
