## [Decision Letter]

Thank you for submitting your article "Prediction error induced motor contagions in human behaviors" for consideration by *eLife*. Your article has been reviewed by three peer reviewers, and the evaluation has been overseen by a Senior Editor and a Reviewing Editor. The following individual involved in review of your submission has agreed to reveal his identity: Kyle Dunovan (Reviewer #3).

The reviewers have discussed the reviews with one another and the Reviewing Editor has drafted this decision to help you prepare a revised submission.

Summary:

We all found the paper very interesting. The experiment was straightforward and the results support the conclusions. However, we have several concerns that will need to be addressed regarding the generality of the findings, presentation of the data, and better highlighting of the novelty of the findings. These points are outlined in detail below.

Essential revisions:

First, whereas we appreciate the ecological approach taken in the study, we are concerned that the findings may only be applicable to experts in particular context (varsity-level baseball players engaging in a pitching task). If this study is revealing a general principle about how prediction errors can be defined through instructions or interpretation of another person's intent, then the findings should replicate in non-experts in a different task. Indeed, previous studies – including your own – have found significant differences between experts and non-experts in perception. A potential worry here is that the varsity-level baseball players have become experts in learning from observation and that these findings will not extend to the general population.

To address this concern, we would like you to conduct a similar study in a group of non-experts in a computerized variant of the task. While we will leave the specific details of this task up to you, one suggestion would be to have two groups of non-experts make reaching or shooting movements to bring a cursor to a target zone after observing errors. One group would be instructed that the intention of the observed behavior was to hit the center of the target (PE group), while another group would be instructed that the intention was to hit a different zone near the target (nPE group). The prediction here is that the PE group will move in a direction to counteract the errors and the nPE group will move in the direction of the error, consistent with your current findings. This follow-up study would go a long way in showing that your findings reveal a general principle about how prediction errors are interpreted.

Second, a peculiar finding in the current study is that both the nPE and PE groups' baseline behavior – prior to observing the errors – are already in the direction of the final result. While we suspect that this just reflects noise in the sample, it might be clearer to understand the results if there were not so heavily binned. We suggest that you re-create Figure 3, but showing all the trials rather than binning each block (including 95% confidence intervals of the mean). Presenting data from individual subjects may also be illuminating. We acknowledge that these results may be messy to look at and, ultimately, may not be the best format for the paper; nonetheless, we would like you to at least include them in the response to the reviewers.

Third, we would like you to analyze the within-subject variability of the data following the observation block. One possibility is that within-subject variance increases due to prolonged absence of visual feedback and that these increases are equal for all three groups (CON, nPE, and PE). An alternative outcome is that the nPE and PE groups are affected by both the direction of the observed throws and the variability of these throws. Here, an interesting outcome is that the variability of the nPE and PE groups in the last session would be similar to the variability in the videos. We think this outcome would strengthen conclusions regarding motor contagion as a direct result of observation.

Finally, previous work by Gribble and colleagues has found that subjects display enhanced adaptation after observing force field learning. The findings from your PE group are consistent with the findings of Gribble et al. To better differentiate your current findings from this prior work, it may be worth discussing how belief regarding the intention of actor, during observation, can result in either motor contagion or internal model adaptation. Further, we think the paper could have a broader impact if you included a section (in the Discussion) about potential neural and/or computational differences between motor contagions and prediction-error-based internal models.

---

## [Author Response]

Essential revisions:First, whereas we appreciate the ecological approach taken in the study, we are concerned that the findings may only be applicable to experts in particular context (varsity-level baseball players engaging in a pitching task). If this study is revealing a general principle about how prediction errors can be defined through instructions or interpretation of another person's intent, then the findings should replicate in non-experts in a different task. Indeed, previous studies – including your own – have found significant differences between experts and non-experts in perception. A potential worry here is that the varsity-level baseball players have become experts in learning from observation and that these findings will not extend to the general population.To address this concern, we would like you to conduct a similar study in a group of non-experts in a computerized variant of the task. While we will leave the specific details of this task up to you, one suggestion would be to have two groups of non-experts make reaching or shooting movements to bring a cursor to a target zone after observing errors. One group would be instructed that the intention of the observed behavior was to hit the center of the target (PE group), while another group would be instructed that the intention was to hit a different zone near the target (nPE group). The prediction here is that the PE group will move in a direction to counteract the errors and the nPE group will move in the direction of the error, consistent with your current findings. This follow-up study would go a long way in showing that your findings reveal a general principle about how prediction errors are interpreted.

Following the reviewers’ suggestion, we conducted an additional experiment with ‘regular’ adult participants. The participants in this ‘Experiment-2’ were asked to make arm reaching movements towards a target line on a computer screen with their right index finger (see Figure 4A). Similar to our throwing experiment (Experiment-1), we used occlusion goggles to prevent feedback of their final finger position, and showed them the video of another individual whose finger pointing was biased in one of two directions, left or right. We then show that, similar to our throwing experiment, the participant’s movements again deviate in opposite directions depending on whether prediction errors were present or not (See Figure 4B). We again observed a significant interaction in the participants’ motor performance changes between nPE, PE, and CON groups. The reproduction of the same effect in such a simple task was pleasantly surprising, and very exciting for us, and hence we thank the reviewers for the suggestion. The details of this Experiment-2 have been explained in the subsections “2) Experiment-2”, “2) Task and apparatus in Experiment-2”, and” Touch location analysis in Experiment-2” in Materials and methods and in the subsection “Experiment 2” in Results.

Second, a peculiar finding in the current study is that both the nPE and PE groups' baseline behavior – prior to observing the errors – are already in the direction of the final result. While we suspect that this just reflects noise in the sample, it might be clearer to understand the results if there were not so heavily binned. We suggest that you re-create Figure 3, but showing all the trials rather than binning each block (including 95% confidence intervals of the mean). Presenting data from individual subjects may also be illuminating. We acknowledge that these results may be messy to look at and, ultimately, may not be the best format for the paper; nonetheless, we would like you to at least include them in the response to the reviewers.

We agree that with the reviewers that there appears to be a bias in the participants’ baseline motor behaviors (first session) between groups in the baseball experiment (Figure 3). Though the one-way ANOVA on the hit locations in the first session showed no significant effect of groups (F(2,27)=3.110, p=0.061), the p-value was marginal. However, looking at the raw data we could see that this marginal p value was probably coincidental. As requested by the reviewers, we show below the data from each individual participant across the sessions (Author response image 1), and the individual 10 throws from each participant in the first session (Author response image 2). As can be observed, the within-participant variability was quite high in each session and there is no visible difference between the groups in the first session. Moreover, the first session in the new reaching experiment (Experiment-2) showed no significant main effect between groups (F(2,27)=0.35, p=0.7032) as observed in Author response image 3.

**Author response image 1. respfig1:** All trial data from all participants in Experiment-1. Different colors and shapes indicate different groups and participants, respectively. Each data point shows a single trial data of a single participant. Error bars show 95% confidence interval of the mean.

**Author response image 2. respfig2:** The ten throw trials by each of all the thirty participants in the first throwing session of Experiment-1. As in Author response image 1, different colors and shapes indicate different groups and participants, respectively.

**Author response image 3. respfig3:** The ten throw trials by each of all the thirty participants in the first reaching session of Experiment-2. As in Author response image 1, different colors and shapes indicate different groups and participants, respectively.

Third, we would like you to analyze the within-subject variability of the data following the observation block. One possibility is that within-subject variance increases due to prolonged absence of visual feedback and that these increases are equal for all three groups (CON, nPE, and PE). An alternative outcome is that the nPE and PE groups are affected by both the direction of the observed throws and the variability of these throws. Here, an interesting outcome is that the variability of the nPE and PE groups in the last session would be similar to the variability in the videos. We think this outcome would strengthen conclusions regarding motor contagion as a direct result of observation.

We agree with the reviewers that the analysis of within-participant variability is important. We had checked and found that there was no substantial change in the variability, but we seem to have missed to report this result in the manuscript, for which we are sorry.

In Experiment-1, the within-participant variability along both the parallel and orthogonal direction does not show any significant change either across the throwing sessions (parallel: F(4,108)=1.934, P=0.109; orthogonal: F(4,108)=0.861, P=0.490) and between the groups (parallel: F(2,27)=0.465, P=0.633; orthogonal: F(2,27)=1.030, P=0.371). In Experiment-2, the within-participant variability along the x axis does not show any significant change across the throwing sessions (F(4,108)=1.155, P=0.335) and between the groups (parallel: F(2,27)=1.090, P=0.351).

This result suggests that the two distinct motor contagions observed in our study, both emerge as an increase in spatial bias rather than variability in the participants’ action outcome. We have added this detail now in the tenth paragraph of the subsection “Experiment-1” and in the last paragraph of the subsection “Experiment 2”.

Finally, previous work by Gribble and colleagues has found that subjects display enhanced adaptation after observing force field learning. The findings from your PE group are consistent with the findings of Gribble et al. To better differentiate your current findings from this prior work, it may be worth discussing how belief regarding the intention of actor, during observation, can result in either motor contagion or internal model adaptation. Further, we think the paper could have a broader impact if you included a section (in the Discussion) about potential neural and/or computational differences between motor contagions and prediction-error-based internal models.

We thank the reviewers for this suggestion. We agree with the reviewers that further discussions on the differences between our motor contagions and the observational motor learning by Gribble et al., and the corresponding computational mechanisms would improve the Discussion.

We believe that the major difference between the results of Gribble et al., and the contagions we observe here are in the part of the motor system affected by the action observation. We believe that the results by Gribble et al. are a consequence of changes in a participant’s ‘inverse model’ or controller, required for learning of a new motor task (arm reaching in a novel force field), as the authors themselves suggest in Brown et al. (2010). On the other hand, we believe the contagions we report here affect an individual’s forward estimation of his own action outcomes, or his ‘forward model’. This belief is based on our recent studies, where we found that the action changes caused by prediction errors during action observation (Ikegami and Ganesh, 2014) are explained by an effect in an individual’s forward model rather than inverse model or controller (Ikegami and Ganesh, 2017).

It therefore seems that observation can affect both the inverse and the forward modelling processes in the brain. However, observation learning studies, unlike our study here, have traditionally utilized tasks where subjects are explicitly required to learn a new force field. Furthermore, unlike the observational motor learning studies, this study systematically manipulates the participants’ belief regarding the observed actions in order to control the direction of prediction errors. The lack of an explicitly required learning and the effect of belief are probably key to whether and to what extent, observed actions affect the inverse, or forward modeling process, or both. However, further studies are required to address this question concretely.

We have added these points to the Discussion (third paragraph).